# Task-level Differentially Private Meta Learning

**Xinyu Zhou**
Department of Computer Science & Engineering
The Ohio State University
`zhou.3542@buckeyemail.osu.edu`

**Raef Bassily**
Department of Computer Science & Engineering and TDAI Institute
The Ohio State University
`bassily.1@osu.edu`

## Abstract

We study the problem of meta-learning with task-level differential privacy. Meta-learning has received increasing attention recently because of its ability to enable fast generalization to new task with small number of data points. However, the training process of meta learning likely involves exchange of task specific information, which may pose privacy risk especially in some privacy-sensitive applications. Therefore, it is important to provide strong privacy guarantees such that the learning process will not reveal any task sensitive information. To this end, existing works have proposed meta learning algorithms with record-level differential privacy, which is not sufficient in many scenarios since it does not protect the aggregated statistics based on the task dataset as a whole. Moreover, the utility guarantees in the prior work are based on assuming the loss function satisfies both smoothness and quadratic growth conditions, which do not necessarily hold in practice. To address these issues, we propose meta learning algorithms with task-level differential privacy; that is, our algorithms protect the entire dataset for each task. In the case when a single meta model is trained, we give both privacy and utility guarantees assuming only that the loss is convex and Lipschitz. Moreover, we propose a new private clustering-based meta-learning algorithm that enables private meta learning of multiple meta models. This can provide significant accuracy gains over the single meta model paradigm, especially when the tasks distribution cannot be well represented by a single meta model. Finally, we conduct several experiments demonstrating the effectiveness of our proposed algorithms.

## 1 Introduction

Meta learning has received increasing attention recently due to its successful applications in a variety of machine learning domains. Meta learning is the process in which existing learning tasks are used to infer a learning rule that enables faster adaptation to new tasks from the same environment. Specifically, meta learning takes in a collection of tasks (datasets) sampled from an unknown distribution. Each task defines a learning problem with respect to an input dataset. The goal of meta learning is to output a meta-learner which can extract the shared knowledge among the tasks (datasets) and generalize well on a newly sampled task (dataset) with a small number of data points. The performance of a meta learner is measured by *transfer risk*, defined as the expected risk of the output of the algorithm w.r.t. a newly sampled task.

Machine learning algorithms typically rely on gradient-based optimization methods. The optimization-based meta learning framework allows such methods to converge to a solution using a small number of samples. This is done by learning either a good parameter initialization or a regularizer. In meta

36th Conference on Neural Information Processing Systems (NeurIPS 2022).

initialization methods like MAML [1] and Reptile [2], the meta leaner obtains a meta initialization $w_0$ such that one or a few gradient updates of $w_0$ produces a good task-specific model for new tasks. Another class of meta learning algorithms are based on an approach known as meta-regularization [3–6] in which a regularizer is first learned based on all tasks datasets, then a task-specific model is obtained for a new task by minimizing the regularized loss. Meta-regularization has received more attention recently due to its formal utility guarantees and computational efficiency. In meta-regularization, the goal is to find a good bias vector $h$ such that a local learner (base learner) can converge to a good task-specific model by minimizing the regularized empirical loss: $\min_{w \in \mathcal{W}} R_Z(w) + \frac{\lambda}{2} \|w - h\|^2$. In [7], the authors formalize the connection between meta-initialization and meta-regularization under the framework of online convex optimization.

Meanwhile, the nature of the meta learning requires the exchange of task-specific information, which may pose serious privacy risks, especially in distributed settings where all tasks (users) are participating into a meta training process by communicating meta-model updates among themselves, typically with the assistance of a central server. Therefore it is important to ensure that the sensitive information in every task (dataset) remains private through the meta learning process.

Despite its importance, the privacy aspect of meta learning is still underexplored. In [8], using the meta-initialization framework, the authors propose a meta-learning algorithm with *record-level* differential privacy (DP) and give an accuracy guarantee in terms of the *transfer risk*. Record-level DP is a weak privacy guarantee for a wide range of scenarios. Imagine training an email spam filter for each user (task). Even if we ensure differential privacy w.r.t. each record of a user's email, aggregate statistics across the entirety of a user's records (e.g., the frequency of certain words or sentences) may still reveal some sensitive information which the user doesn't want to share. Moreover, the accuracy guarantees in [8] rely on relatively strong assumptions concerning the loss function used for training. In particular, the loss function therein is assumed to satisfy both smoothness and quadratic growth conditions, which do not hold simultaneously for many loss functions commonly used in practice.

In this work, we propose new meta-learning algorithms with task-level DP and derive formal transfer risk bounds. Unlike [8], our algorithms are based on the meta-regularization framework. We now highlight our main technical contributions:

- We propose a task-level differentially private (task-level DP) meta learning algorithm based on the meta-regularization approach. We observe that when the meta-loss is properly defined, one can view each task dataset as a "data point" and the meta-loss as its corresponding loss function. Therefore, applying existing DP optimization techniques like Noisy SGD on the newly defined "data point" and loss function achieves task-level differential privacy naturally.

- We give a bound for the statistical performance for our algorithm in terms of its excess transfer risk. Our statistical accuracy guarantee holds under standard assumptions; namely, assuming only that loss function is convex and Lipshitz. With sufficiently large number of tasks, the bound asserts improved performance over local training, where a model is trained locally based on a single task dataset.

- When the task distribution becomes more complex such that a single meta model is not enough to capture the relationship among tasks, we propose a task-level DP clustering-based meta learning algorithm that can learn multiple meta models. This approach can provide significant improvements in the statistical accuracy (transfer risk) over the standard single-model case, especially when the tasks distribution is better captured by multiple meta models.

- We empirically evaluate our algorithms to demonstrate the effectiveness of our proposed algorithms[1].

**Related work:** The most related work to ours is [8], where the authors propose a meta-learning algorithm with record-level DP via the framework of meta-initialization. Their analysis relies on assuming that the loss function satisfies a quadratic growth condition, which we do not require in our analysis. Furthermore, we consider a stronger notion of the excess transfer risk than the one in [8]. To be more specific, in [8], the achieved transfer risk is compared to the transfer risk of the optimal *global* model, while the transfer risk in our work is compared to that of the optimal *personalized* model. This is because we allow different tasks to have different parameters. Since

---

[1]Our code is available online at `https://github.com/xyzhou055/MetaNSGD`

the optimal personalized model always outperforms global model, we have a stronger notion of the excess transfer risk. Consequently, our utility guarantee also holds under the notion studied in [8].

Our results are also related to the line of work in private model personalization. The goal of model personalization is to train a personalized model for each user that generalizes well on the user's distribution. Reference [9] proposes a private personalization algorithm that solves a stochastic optimization problem involving both local and global models. The authors of [10] develop an algorithm for mean-regularized multi-task learning under the joint differential privacy guarantee and study its application in model personalization. The work of [11] presents private personalization algorithms that learn a common embedding model among users using the alternating minimization framework, where convergence to the optimal solution is guaranteed for linear regression models with Gaussian data. There are also a variety of works, mainly in the non-private regime, exploring the connection between meta learning and model personalization. For example, [12] proposes a personalization mechanism based on MAML, which aims at finding a good initialization parameter for the users. The authors of [13] study the connection between meta learning and personalization based on the observation that FedAvg [14] can be seen as a meta learning algorithm. Therefore, it is possible to extend the results in this work to the domain of model personalization.

There are also prior works studying learning with task-level or user-level differential privacy but under different learning paradigms. In [15, 16], the authors provide user-level differential privacy under the framework of federated learning, in which case all users share a single global model. This is different from our setting because we allow each user to have its own task-specific model. The authors in [17] develop algorithms for a variety of learning tasks under user-level differential privacy. They consider a limited heterogeneity setting where all users' distributions are close to each other in total variation distance. On the other hand, in this work we focus on the meta learning paradigm, where we only assume small variation across users' models.

## 2 Preliminaries

**Distributional setup:** We consider a meta distribution $\rho$ and each task is sampled from $\rho$ which will induce a task specific distribution $\mu$ over the data domain $\mathcal{Z}$, that is $\mu_1, \ldots \mu_K \overset{\text{i.i.d}}{\sim} \rho$ for $K$ tasks. After the sampling of $\mu_i$, a dataset $Z_i$ with $n$ datapoints is subsequently sampled from $\mu_i$ written as $Z_i = \{z_1, \ldots z_n\} \overset{\text{i.i.d}}{\sim} \mu_i^n$. The collection of $K$ task datasets is denoted as

$$\bar{Z} = \{Z_1, \ldots Z_K\}$$

In this paper, we focus on distributed settings such that the datasets $\{Z_i\}_{i=1}^{K}$ are decentralized across multiple locations. All tasks jointly participate into a model training process coordinated by a trusted central server. The data of each task is stored and processed locally, only the model updates are shared to the server or other tasks.

**Task population and empirical losses:** Given a closed convex parameter space $\mathcal{W}$, we consider learning task parametrized by vector $w \in \mathbb{R}^d$. We let $\ell : \mathbb{R}^d \times \mathcal{Z} \to \mathbb{R}$ be a loss function, where $\ell(w, z)$ is the loss incurred by a parameter vector $w$ on a data point $z$. For any task drawn from a distribution $\mu$ and observed via a dataset $Z = (z_1, \ldots z_n)$, we denote the population risk of $w \in \mathcal{W}$ as $R_\mu(w) = \mathbf{E}_{z \sim \mu} \ell(w, z)$ and the empirical risk as $R_Z(w) = \frac{1}{n} \sum_{i=1}^{n} \ell(w, z_i)$.

**Meta learning**: Given $K$ learning tasks with datasets $\bar{Z}$, a meta-learning process aims to learn an algorithm $\mathcal{A}$ that enables fast adaption on newly sampled tasks from the meta-distribution $\rho$. Such adaptation is measured by the notion of *transfer risk* defined as

$$\mathcal{E}_n(\mathcal{A}; \rho) = \mathbf{E}_{\mu \sim \rho} \mathbf{E}_{Z \sim \mu^n} R_\mu(\mathcal{A}(Z)). \tag{1}$$

The transfer risk $\mathcal{E}_n(\mathcal{A}; \rho)$ can be considered as the expected loss of the parameter returned by applying the algorithm $\mathcal{A}$ on a dataset $Z \in \mathcal{Z}^n$ sampled from the meta distribution $\rho$. In summary, the whole process starts with a meta-learning algorithm taking $\bar{Z}$ as input and outputting a learning algorithm $\mathcal{A}$. After that, new learning task with distribution $\mu \sim \rho$ trains its corresponding dataset with the learned algorithm $\mathcal{A}$ and apply the resulting parameter vector $w$ to new data drawn from $\mu$.

For any task specific distribution $\mu$, we denote the minimizer $w_\mu$ as $w_\mu \in \min_{w \in \mathcal{W}} R_\mu(w)$ and the expected minimum error over the meta-distribution $\rho$ as $\mathcal{E}_\rho = \mathbf{E}_{\mu \sim \rho} R_\mu(w_\mu)$. Note that, there might be more than one minimizers of $R_\mu(w)$, but $\mathcal{E}_\rho$ remains same regardless of the selection of $w_\mu$. Furthermore, we define the excess transfer risk as

$$\Delta \mathcal{E}_n(\mathcal{A}; \rho) = \mathcal{E}_n(\mathcal{A}; \rho) - \mathcal{E}_\rho \tag{2}$$

**Task similarity metric:** To achieve low transfer risk, the meta learning algorithm leverages the shared similarity among tasks and output the optimized inner algorithm. In this work, the task similarity is measured by the quantity:

$$V^2 = \min_{\phi \in \mathcal{W}} \mathbf{E}_{\mu \sim \rho} \left\| \phi - \text{Proj}_{W_\mu}(\phi) \right\|^2 \tag{3}$$

where $W_\mu$ is the set of optimal parameters for task $\mu$ written as $W_\mu = \arg\min_{w \in \mathcal{W}} R_\mu(w)$, and $\text{Proj}_{W_\mu}$ denotes the Euclidean projection on $W_\mu$. $V$ can be roughly seen as the expected variation of task parameters. We denote $\bar{w}$ as the minimizer of (3) such that $\bar{w} \in \mathcal{W}$ and $V^2 = \left\| \bar{w} - \text{Proj}_{W_\mu}(\bar{w}) \right\|^2$.

**Task-level Differential Privacy** [18]: A randomized algorithm $\mathcal{M}$ is $(\epsilon, \delta)$-differential private if for any pair of neighboring sets $\bar{Z}$ and $\bar{Z}'$ and for any measurable subset $\mathcal{O}$ of $\text{Range}(\mathcal{M})$, we have

$$P(\mathcal{M}(\bar{Z}) \in \mathcal{O}) \le e^\epsilon P(\mathcal{M}(\bar{Z}') \in \mathcal{O}) + \delta$$

In this work, we consider task-level neighboring relationship. Let $\bar{Z} = \{Z_1, \dots Z_K\}$ and $\bar{Z}' = \{Z_1', \dots Z_K'\}$, and we say $\bar{Z}$ and $\bar{Z}'$ are neighboring if, for some $i \in [K]$, $\bar{Z}$ and $\bar{Z}'$ differ only on the dataset from the $i_{th}$ task, i.e., $Z_j = Z_j'$ for all $j$ except $i$. Therefore, a task-level differential private algorithm ensures that any adversary cannot tell if the dataset of a particular task was used in the algorithm given its output.

Our analysis involves the following fairly standard assumption:

**Assumption 1.** *(Convex-Lipschitz-Bounded models)*

- *For any $z \in \mathcal{Z}$, the loss function $\ell(w, z)$ is convex and $L$-Lipshitz with respect to $w$.*
- *Bounded parameter space: $\|\mathcal{W}\|_2 = \max_{w_1, w_2 \in \mathcal{W}} \|w_1 - w_2\| \le M$*

## 3 Learning with Meta Regularization

**Base learner:** A base learner solves a regularized loss minimization problem w.r.t. a biased regularizer parameterized by a bias vector $h$. In particular, given a bias vector $h$ and a task dataset $Z$, the base learner solves the following problem:

$$\min_{w \in \mathcal{W}} R_{Z,\lambda}(w; h) = \min_{w \in \mathcal{W}} R_Z(w) + \frac{\lambda}{2} \|w - h\|^2$$

where $\lambda$ is the regularization parameter. In the sequel, we define $w_h(Z)$ as the minimizer of $R_{Z,\lambda}(w; h)$, that is $w_h(Z) = \arg\min_{w \in \mathcal{W}} R_{Z,\lambda}(w; h)$.

**Meta algorithm:** Given a collection of base learners, the goal of a meta-algorithm is to find a good regularization bias $h$ that yields a small transfer risk. First, let's rewrite the transfer risk defined in (1) as a function $h$:

$$\mathcal{E}_n(h; \rho) = \mathbf{E}_{\mu \sim \rho} \mathbf{E}_{Z \sim \mu^n} R_\mu(w_h(Z)) \tag{4}$$

and hence the excess transfer risk can be expressed as $\Delta \mathcal{E}_n(h; \rho) = \mathcal{E}_n(h; \rho) - \mathcal{E}_\rho$. Therefore, the goal of the meta algorithm is to find an (approximate) minimizer $h^*$ of $\mathcal{E}_n(h; \rho)$ in (4), i.e., the goal is to solve:

$$\min_{h \in \mathcal{W}} \mathcal{E}_n(h; \rho) = \mathbf{E}_{\mu \sim \rho} \mathbf{E}_{Z \sim \mu^n} R_\mu(w_h(Z)) \tag{5}$$

Intuitively speaking, the base learner aims to figure out the best parameter vector "around" the bias vector $h$. Therefore, if the meta algorithm can find a good bias vector $h$ that is close to the solution space of the tasks, the base learner can converge to its solution space quickly by only searching around $h$.

### 3.1 Meta Loss and proxy transfer risk

Directly optimizing over (5) above is usually infeasible. A common strategy to get around this difficulty is to propose a proxy loss function which is related to the transfer risk but easier to handle. In this section, we follow the definitions in [3, 4] and define our meta loss as

$$L_Z(h) = \min_{w \in \mathcal{W}} R_{Z,\lambda}(w; h) = \min_{w \in \mathcal{W}} R_Z(w) + \frac{\lambda}{2} \|w - h\|^2$$

and the proxy to the transfer risk is defined as

$$\hat{\mathcal{E}}_n(h;\rho) = \mathbf{E}_{\mu \sim \rho} \mathbf{E}_{Z \sim \mu^n} L_Z(h)$$

The goal of the meta-algorithm is to optimize the proxy risk over $h$, that is,

$$\min_{h \in \mathcal{W}} \hat{\mathcal{E}}_n(h;\rho) = \mathbf{E}_{\mu \sim \rho} \mathbf{E}_{Z \sim \mu^n} L_Z(h) \tag{6}$$

Note that $L_Z(h)$ is in fact the Moreau envelope of the empirical loss $R_Z(w)$. In Proposition 2, we review some useful properties of $L_Z(h)$ which will help our analysis.

**Proposition 2.** *(follows from [19, 20]) Under Assumption 1, we have*

1. *$L_Z$ is convex, $2L$-Lipshitz and $\lambda$-smooth.*

2. *For any $h \in \mathcal{W}$, $\nabla L_Z(h) = -\lambda(w_h(Z) - h)$ where $w_h(Z) = \arg\min_{w \in \mathcal{W}} R_{Z,\lambda}(w;h)$.*

A key observation is that all datasets go through the same stochastic process under our distributional setup and thus can be seen as i.i.d. sequence drawn from a distribution $P_Z$. Therefore, (6) can be rewritten as

$$\min_{h \in \mathcal{W}} \hat{\mathcal{E}}_n(h;\rho) = \min_{h \in \mathcal{W}} \mathbf{E}_{Z \sim P_Z} L_Z(h) \tag{7}$$

Combined with Proposition (2), the objective of the meta algorithm is formulated as a stochastic convex optimization (SCO) problem which has been well studied under privacy constraints [19, 21, 22]. The notion of task-level differential privacy just comes up naturally when we see each task dataset $Z$ as a "data point" and $L_Z(h)$ as its loss function, therefore the conventional record-level neighboring relationship can be readily translated to a task-level neighboring relationship.

However, one challenge in adapting existing differentially private SCO methods pertains to the gradient computation. In standard SCO settings, we are expected to be able to compute the gradient of the loss function efficiently and precisely. However, in our case, as seen in Proposition 2, computing $\nabla L_Z(h)$ for each $h$ requires computing $w_h(Z)$ which further involves solving a $\lambda$-strongly convex optimization problem. Therefore, finding the exact value of $w_h(Z)$ for general convex loss $\ell$ is not feasible, alternatively what we can do is to approximate $w_h(Z)$ using some iterative methods up to some error $b$. Therefore, it is important to incorporate this approximation error into our convergence analysis. Moreover, due to the distributed nature of the algorithm, the model iterates are assumed to be visible to multiple tasks who are selected to participate in each round. Therefore, the privacy should be defined over all model iterates as well as the final model, and we cannot use the linear-time algorithm for DP smooth stochastic optimization [22] despite the fact that the objective of the meta-algorithm can be formulated as a stochastic convex optimization problem with a smooth loss.

## 4 Private Meta Learning with Noisy SGD

In this section, we present our meta learning algorithm adapted from the mini batch noisy SGD method in [19] described in Algorithm 1. In iteration $t$, given the current iterate $h_t$, each sampled task computes an approximation of the gradient of the meta loss $L_Z(h_t)$, denoted as $\hat{\nabla} L_Z(h_t)$. After that, the server aggregates theses approximate gradients to update the model. The privacy is achieved by adding Gaussian noise scaled with the $l_2$ sensitivity of each task to the model iterate.

We will first describe the privacy and utility guarantee of Algorithm 1, and then show the resulting transfer risk.

**Theorem 3.** *(Privacy guarantee) Let Assumption 1 holds and let $\{h_1, \ldots h_{T-1}\}$ and $\bar{h}_T$ be the output, Algorithm 1 is $(\epsilon, \delta)$-differentially private on task level.*

*Proof.* Since the computed gradient $\hat{\nabla} L_Z(h_t)$ is only an approximation to $\nabla L_Z(h_t)$, then we have

$$\left\| \hat{\nabla} L_Z(h_t) - \nabla L_Z(h_t) \right\| \le \lambda \left\| \hat{w}_{h_t}(Z) - w_{h_t}(Z) \right\| \le b$$

and because $\nabla L_Z(h)$ is $2L$-Lipshitz by Proposition 2, we have

$$\left\| \hat{\nabla} L_Z(h_t) \right\| \le \left\| \nabla L_Z(h_t) \right\| + b \le 2L + b$$

---

**Algorithm 1:** $\mathcal{A}_{\text{meta-NSGD}}$: meta learning with mini-batch noisy SGD

---

**input** : Initial iterate $w_0 \in \mathcal{W}$, number of iterations $T$, estimation error $b$, regularization
parameter $\lambda$, step size $\eta$.

1 Set noise variance $\sigma^2 = \frac{8T(2L+b)^2 \log(1/\delta)}{K^2 \epsilon^2}$ and batch size $m = \max\left(K\sqrt{\epsilon/(4T)}, 1\right)$

2 **for** $t = 1, \ldots, T-1$ **do**

3     Server samples a batch of tasks $B_t = \{Z_{i_{(t,1)}} \ldots Z_{i_{(t,m)}}\} \leftarrow \bar{Z}$ and sends $h_t$ to tasks in $B_t$

4     **for** each task $Z$ in $B_t$ **do**

5        Solve $R_{Z,\lambda}(w; h_t)$ and compute an estimate $\hat{w}_{h_t}(Z)$ with $\|\hat{w}_{h_t}(Z) - w_{h_t}(Z)\| \leq \frac{b}{\lambda}$

6        Send $\hat{\nabla}L_Z(h_t) = -\lambda(\hat{w}_{h_t}(Z) - h_t)$ back to server

7     **end**

8     Server updates the model with

$$h_{t+1} = \text{Proj}_{\mathcal{W}}\left(h_t - \eta \cdot \left(\frac{1}{m}\sum_{Z \in B_t} \hat{\nabla}L_Z(h_t) + G_t\right)\right)$$

    where $\text{Proj}_{\mathcal{W}}$ denotes the Euclidean projection onto $\mathcal{W}$, and $G_t \sim \mathcal{N}(0, \sigma^2 \mathbb{I}_d)$ drawn
    independently each iteration.

9 **end**

**output** : $\bar{h}_T = \frac{1}{T}\sum_{t=1}^{T} h_t$

---

Note that Algorithm 1 has the similar structure of the mini-batch noisy SGD method in [19] if we consider $\hat{\nabla}L_Z(h_t)$ as the model updates and $h_t$ as model iterates. By adding Gaussian noise scaled with $(2L + b)$, a direct application of Theorem 3.1 in [19] completes the proof. $\qquad\square$

Since the objective of the base learner $R_{Z,\lambda}(w; h_t)$ is a $\lambda$-strongly convex problem, one can use gradient decent to bound the approximation error at step 5 in Algorithm 1. Alternatively, the base learner can use other stochastic optimization method i.e. SGD to achieve a high probability bound on the approximation error, such that $\|\hat{w}_{h_t}(Z) - w_{h_t}(Z)\| \leq \frac{b}{\lambda}$ with probability over $1 - \delta'$. By letting the overall failure probability $mT\delta'$ on the same order of the privacy parameter $\delta$, the privacy guarantee still holds. For the simplicity of the analysis, we assume the approximation error $\|\hat{w}_{h_t}(Z) - w_{h_t}(Z)\| \leq \frac{b}{\lambda}$ always satisfies.

In the next theorem, we will give the excess risk of $\mathcal{A}_{\text{meta-NSGD}}$

**Theorem 4.** *(Utility guarantee) Under Assumptions 1, suppose $\lambda \leq \frac{2L}{M} \cdot \min\left(\sqrt{\frac{K}{2}}, \frac{\epsilon K}{2\sqrt{2d\log(1/\delta)}}\right)$.*
*Let $T = \min\left(\frac{K}{8}, \frac{\epsilon^2 K^2}{32d\log(1/\delta)}\right)$, $\eta = \frac{M}{2L\sqrt{T}}$ and $b = \frac{2L}{K}$. Then,*

$$\mathbf{E}[\hat{\mathcal{E}}_n(\bar{h}_T; \rho)] - \min_{h \in \mathcal{W}} \hat{\mathcal{E}}_n(h; \rho) \leq O\left(ML \cdot \left(\frac{1}{\sqrt{K}} + \frac{\sqrt{d\log(1/\delta)}}{\epsilon K}\right)\right)$$

*where $\bar{h}_T$ is the final model output by Algorithm 1, and the expectation is taken over the collection of datasets $\bar{Z}$ and the inner randomness of the algorithm.*

Note that the excess rate in Theorem 4 is with respect to $\hat{\mathcal{E}}_n(h; \rho)$, the proxy to the transfer risk. It remains to show how it can be related to the excess rate of the true transfer risk $\mathcal{E}_n(h; \rho)$. In the next theorem, we give the excess transfer risk of Algorithm 1. The proof follows from [Theorem 6, [3]] and decomposes the excess transfer rate into three components. We bound each component separately and one of them is based on the excess rate achieved in Theorem 4.

**Theorem 5.** *(Transfer risk) Recall the settings in Theorem 4, we have*

$$\mathbf{E}[\mathcal{E}_n(\bar{h}_T; \rho)] - \mathcal{E}_\rho \leq \frac{2L^2}{\lambda n} + \frac{\lambda}{2}V^2 + O\left(ML\left(\frac{\sqrt{d\log(1/\delta)}}{\epsilon K} + \frac{1}{\sqrt{K}}\right)\right)$$

*By letting $\lambda = \frac{2L}{V}\sqrt{\frac{1}{n}}$, we have*

$$\mathbf{E}[\mathcal{E}_n(\bar{h}_T; \rho)] - \mathcal{E}_\rho \leq \frac{2LV}{\sqrt{n}} + O\left(ML \cdot \left(\frac{\sqrt{d\log(1/\delta)}}{\epsilon K} + \frac{1}{\sqrt{K}}\right)\right)$$

*where $\bar{h}_T$ is the output of $\mathcal{A}_{meta\text{-}NSGD}$ and the expectation is taken w.r.t. $\bar{Z}$ and the inner randomness of the algorithm.*

Theorem 5 shows that when the number of tasks $K$ is sufficiently large, the excess transfer risk becomes $O\left(\frac{LV}{\sqrt{n}}\right)$. Recall that $V$ is a measure of task similarity, hence when the optimal task parameters are close to each other, $V$ can be much smaller than the diameter of the parameter space which indicates better performance than local training where each task trains its own model independently. Note one can also give a direct bound of the transfer risk of the output of Algorithm 1 based on the fact that $\left\|\bar{h}_T - w_\mu\right\| \leq \|\mathcal{W}\| = M$ for any task $\mu$ with optimal parameter $w_\mu$, standard derivation gives that $\mathbf{E}[\mathcal{E}_n(\bar{h}_T; \rho)] - \mathcal{E}_\rho \leq O\left(\frac{LM}{\sqrt{n}}\right)$ which is the same as the excess risk for local training. Therefore, Algorithm 1 is at least as good as local training, in the worst case, it recover the performance of local training.

The rate achieved in Theorem 5 is on the same order of the non-private rate $O\left(\frac{VL}{\sqrt{n}} + \frac{ML}{\sqrt{K}}\right)$ from [3] in the regime most common in practice where $d = O(K)$ and $\epsilon = \theta(1)$. Note that [8] obtains $O\left(\frac{1}{K}\right)$ convergence on the number of tasks $K$ under record-level privacy guarantee, but their analysis relies on the assumption for the loss function to be $\alpha$-quadratic growth, which we do not require for our analysis. Furthermore, we have different definitions of excess transfer risk from [8], given the meta learner $\mathcal{A}_h$, the excess transfer risk in [8] is defined as $\mathcal{E}_n(\mathcal{A}_h; \rho) - \min_{w \in \mathcal{W}} \mathbf{E}_{\mu \sim \rho} R_\mu(w)$ while ours is defined as $\mathcal{E}_n(\mathcal{A}_h; \rho) - \mathbf{E}_{\mu \sim \rho}[\min_{w \in \mathcal{W}} R_\mu(w)]$. Since $\min_{w \in \mathcal{W}} \mathbf{E}_{\mu \sim \rho} R_\mu(w) \geq \mathbf{E}_{\mu \sim \rho}[\min_{w \in \mathcal{W}} R_\mu(w)]$, our bound holds under a stronger definition for the excess transfer risk. Hence, our bound is a valid upper bound for the notion defined in [8]. The detailed proofs of Theorems 4 and 5 can be found in the appendix.

**Communication-efficient, private meta Learning with DP-FTMRL**    Although Algorithm 1 gives a transfer risk comparable with the non-private rate, it requires $O\left(\min\left\{\sqrt{K}, K^{3/2}/d\right\}\right)$ rounds of communication to each task on average. This can be expensive when the task number $K$ is large but each task only has limited computation and communication resource, which is a common setting in distributed scenarios like federated learning. Therefore, we introduce a communication efficient DP meta algorithm Differentially Private Follow-the-Meta-Regularized-Leader (DP-FTMRL) which is based on the DP-FTRL algorithm in [23]. DP-FTMRL is a one-pass algorithm which means that every task will be communicated by the server for only once. Furthermore, it processes each task sequentially thus allows tasks to arrive in a stream. With high probability over the algorithm inner randomness, DP-FTMRL obtains transfer risk of $\tilde{O}\left(\frac{LV}{\sqrt{n}} + ML\left(\frac{1}{\sqrt{K}} + \frac{d^{1/4}}{\sqrt{K}\epsilon}\right)\right)$. This rate is suboptimal compared with the one achieved in Theorem 5, but DP-FTMRL provides significant savings on the communication cost. More details of DP-FTMRL can be found in the appendix.

**Note:**    To get the excess transfer risk guarantee in Theorem 5, we need to set $\lambda = \frac{2L}{V}\frac{1}{\sqrt{n}}$ which requires the knowledge of the value of $V$, the task similarity. In fact, we can privately approximate $V$ using its empirical counterpart; in particular, we can find a private counterpart for $\hat{V}^2 \triangleq \min_{\phi \in \mathcal{W}} \frac{1}{K}\sum_{i=1}^{K} \|\phi - \hat{w}(Z_i)\|^2$ where $\hat{w}(Z_i)$ is a minimizer of the empirical loss of task $Z_i$. This can be done using standard DP convex optimization methods such Noisy Gradient Descent. An alternative approach to choose a setting for $\lambda$, which is quite common in practice, is to treat $\lambda$ as a hyperparameter [4] and tune it privately. This can also be done using existing methods in the literature of DP [24, 25].

## 5    Private Clustering-Based Meta Learning

In this section, we consider the case where the tasks are sampled from distribution with multiple concentrations (e.g., a multi-modal distribution) where a single meta model is not enough to capture the relationship among tasks. A natural extension is to cluster the tasks into several groups and train a meta model for each group.

To provide a more formal treatment, we let $q$ denote the number of possible meta models (clusters), each is parametrized with a bias vector $h_i$. These meta models are denoted as $(h_1, h_2, \ldots h_q)$. The task similarity in the presence of $q$ meta modals is hence modified to

$$V_c^2 = \min_{h_1, \ldots h_q \in \mathcal{W}} \mathbf{E}_{\mu \sim \rho} \min_{i \in [q]} \left\| h_i - \text{Proj}_{W_\mu}(h_i) \right\|^2 \tag{8}$$

Recall that $W_\mu$ is the set of optimal parameters for task $\mu$. Note that $V_c$ is no larger than $V$ from single meta learner case defined in (3). In the case when $\{W_\mu\}$ contains multiple disjoint and far apart concentrations, $V_c$ can be much smaller compared with $V$.

In particular, inspired by the idea of hypothesis based clustering from [26], we define the objective of the meta-algorithm with $q$ meta learners as

$$\min_{h_1, \ldots h_q \in \mathcal{W}} \hat{\mathcal{E}}_n(\mathbf{h}, \bar{Z})) = \frac{1}{K} \sum_{j=1}^{K} \min_{i \in [q]} L_Z(h_i) \tag{9}$$

where $L_Z(\cdot)$ is the meta loss.

Given $\mathbf{h} = (h_1 \ldots h_q)$, the base learner with dataset set $Z$ first chooses the $h_i$ with lowest meta loss, that is, $C(Z) = \arg\min_{i \in [q]} L_Z(h_i)$ and output the corresponding minimizer $w_{h_{C(Z)}}(Z) = \min_{w \in \mathcal{W}} R_{Z,\lambda}(w; h_{C(Z)}) = \min_{w \in \mathcal{W}} R_Z(w) + \frac{\lambda}{2} \left\| w - h_{C(Z)} \right\|^2$. Therefore, the transfer risk is defined as

$$\mathcal{E}_n(\mathbf{h}; \rho) = \mathbf{E}_{\mu \sim \rho} \mathbf{E}_{Z \sim \mu^n} R_\mu \left( w_{h_{C(Z)}(Z)} \right)$$

We present our meta-clustering algorithm described in Algorithm 2, which is inspired by the HYP-CLUSTER algorithm in [26]. Algorithm 2 is a heuristic algorithm aiming at minimizing the objective (9) over $h_1, \ldots h_q$. Each iteration consists of two steps, the model selection step and model update step. At the beginning of one iteration, each sampled task selects the model from $\{h_i\}_{i=1}^{q}$ that incurs the lowest loss. After the assignment is done, the server will update each model with its corresponding tasks.

We formally show the privacy guarantee of Algorithm 2. The performance of this algorithm in terms of its transfer risk will be evaluated empirically since convergence to a true optima is not guaranteed due to the non-convexity of $\hat{\mathcal{E}}_n(\mathbf{h}, \bar{Z}))$ in $\mathbf{h}$.

---

**Algorithm 2:** $\mathcal{A}_{\text{meta-cluster}}$: private clustering-based meta learning algorithm

**input** : $q$ randomly initialized iterates $h_0^1, h_0^2, \ldots h_0^q$, number of iterations $T$, batch size $m$

1 Set noise variance $\sigma^2 = \frac{8T(2L+b)^2 \log(1/\delta)}{K^2 \epsilon^2}$

2 **for** $t = 1, \ldots, T-1$ **do**

3      Server samples a batch of tasks $B_t$ of size $m$ and sends $\{h_t^i\}_{i=1}^q$ to $B_t$

4      **for** *each task $Z$ in $B_t$* **do**

5          For $i \in [q]$, compute approximate minimizers of $R_{Z,\lambda}(w, h_t^i)$ as $\hat{w}_h^i(Z)$ with approximation error $\frac{b}{\lambda}$

6          Select the model with smallest meta loss $C(Z) = \arg\min_{i \in [q]} R_{Z,\lambda}(\hat{w}_h^i(Z), h_t^i)$

7          Compute the gradient $\hat{\nabla} L_Z(h_t^{C(Z)}) = -\lambda \left( \hat{w}_Z^{C(Z)} - h_t^{C(Z)} \right)$

8          Send $(C(Z), \hat{\nabla} L_Z(h_t^{C(Z)}))$ back to server

9      **end**

10      For every $i \in [q]$, denote the set $C_t^i = \{Z \in B_t | C(Z) = i\}$, server updates $h_t^i$ as

$$h_{t+1}^i = \text{Proj}_{\mathcal{W}} \left( h_t^i - \eta \cdot \left( \frac{1}{m} \sum_{Z \in C_t^i} \hat{\nabla} L_Z(h_t^i) + G_t^i \right) \right)$$

     where $G_t^i \sim \mathcal{N}(0, \sigma^2 \mathbb{I}_d)$ drawn independently each iteration $t$ and meta model $i$.

11 **end**

     **output :** $\{h_T^1, h_T^2, \ldots h_T^q\}$

---

**Theorem 6.** *(Privacy guarantee) Let $\{h_t^1, \ldots h_t^q\}_{t \in T}$ be the output, Algorithm 2 is $(\epsilon, \delta)$-differentially private on task level.*

*Proof.* Similar to Theorem 3, we have the norm of $\hat{\nabla} L_Z(h_t^{C(Z)})$ is bounded by $(2L + b)$ for any choice of $C(Z)$. Since each task only selects one meta model to update in each iteration, we consider $\{h_t^i\}_{i=1}^q$ as the model iterate and $\hat{\nabla} L_Z(h_t^i)$ as the model update from the task, we can use the same argument in Theorem 3 to complete the proof. $\square$

## 6 Experiments

In this section, we empirically evaluate the performance of our algorithms under various privacy regimes. On the one hand, we demonstrate the performance improvement of our algorithms over local training. Meanwhile, when the tasks are sampled from multi-modal distribution, we show that our clustering based algorithm achieves lower transfer risk than the single meta model paradigm.

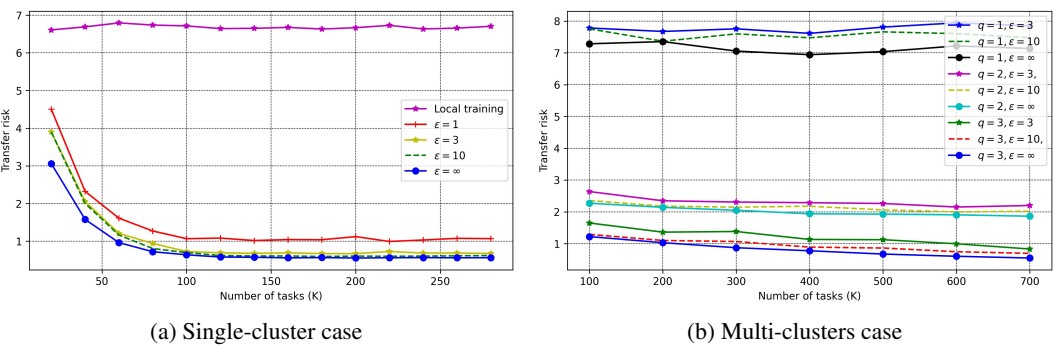

(a) Single-cluster case         (b) Multi-clusters case

Figure 1: The transfer risk of $\mathcal{A}_{\text{meta-cluster}}$ in linear regression task with respect to the number of tasks. $\epsilon$ is the privacy parameter and $q$ is the number of meta models (clusters).

We consider linear regression task with mean square loss. The weight vector $w_\mu$ is $d$-dimentional vectors that concentrates around one or more common vectors. We let $d = 30$ for all experiments. The feature vector $x$ is uniformly sampled from unit ball, and the label $y = \langle w_u, x \rangle + n$ where $n$ is the zero mean Gaussian noise with standard deviation equal to $0.5$.

**Single cluster case:** For each task $\mu$, its $d$-dimensional weight vector $w_\mu$ is sampled from Gaussian distribution with mean $\bar{h}$ and with variance $\sigma^2$, denoted as $\mathcal{N}(\bar{h}, \sigma^2 \mathbb{I}_d)$. In the experiment, $\sigma = 1$ and $\bar{h}$ is a vector with all entries equal to $4$. Each task contains 10 datapoints $(x_i, y_i)_{i=1}^{10}$. The privacy parameters $\epsilon$ are chosen from $\{1, 3, 10\}$ and $\delta$ is set as $10^{-5}$. We set the clipping norm to 2 and use the functions from *TensorFlow privacy*[27] to choose the noise multiplier to fit the privacy budget. In Figure 1, the transfer risks of our meta learning algorithm are substantially better than the transfer risk of local training even in high privacy regimes when $\epsilon = 1$. Furthermore, by comparing the transfer risks for different $\epsilon$'s, we show that the performance gaps are much less significant compared with the gap between local training. In modest privacy regimes when $\epsilon = 3$ or $10$, the corresponding transfer risks are quite close to the non-private rate ($\epsilon = \infty$).

**Multi clusters case:** The weight vector is sampled from the distribution $\mathcal{N}(\bar{h}, \sigma^2 \mathcal{I}_d)$ where $\bar{h}$ is uniformly sampled from the set $\mathbf{h} = \{\bar{h}_1, \ldots \bar{h}_t\}$. In the experiment, $t$, the number of cluster in the underlying distribution, is set to be 3, $\sigma = 0.5$ and we choose $\{\bar{h}_1, \bar{h}_2, \bar{h}_3\}$ to be three orthogonal vectors with different norms. More specifically, $\bar{h}_1 = [2 \ldots 2, 0, \ldots 0]$, $\bar{h}_2 = [0, \ldots, 0, -4 \ldots, -4, 0, \ldots 0]$ and $\bar{h}_3 = [0, \ldots, 0, 6, \ldots 6]$, and each $\bar{w}_i$ has 10 non-zero components. The privacy parameters $\epsilon$ are chosen from $\{3, 10\}$ and $\delta$ is set as $10^{-5}$, the clipping norm is set to 1. Furthermore, we set $q$, the number of meta models in Algorithm 2, from 1 to 3 and report the transfer risk with each $q$. The plots in Figure 2 confirms our claim that we can achieve better performance by adding the number of meta models. By choosing different number of meta models, the performance gap can be significant. Meanwhile, similar to the single cluster case, with the same number of clusters, we see relatively small performance difference under various privacy parameters.

In Appendix C, we present additional experiments to evaluate our algorithms on Omniglot [28] few-shot classification tasks, where we demonstrate an improved performance over existing DP meta learning methods.

## 7 Conclusion and Future Work

In this work, we studied the meta learning problem with task-level privacy. When a single meta model is trained, we have shown, both theoretically and empirically, that our private meta algorithm achieves better performance than local training when the variation among the tasks is relatively small. To deal with more complex task distribution, we also proposed a private clustering-based meta learning algorithm that outputs multiple meta models. This approach can provide noticeable performance gains over the single meta model paradigm as demonstrated by our empirical results.

An interesting direction for future work is to formally study the accuracy guarantees of our clustering-based algorithm. Due to the non-convexity of the objective function, theoretical analysis on the transfer risk may be challenging. However, it is still possible to obtain a guarantee to characterize the generalization property of the clustering-based algorithm.

## Acknowledgments and Disclosure of Funding

This work is supported by NSF AI-Institute Award 2112471.

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
