# A Proof of results in Section 4

## A.1 Auxiliary lemmas

Given a function $f$ which is convex, $L$-Lipschitz and $\beta$-smooth. Define the update rule $G_{f,\eta}$ as

$$G_{f,\eta}(w) = w - \eta \nabla f(w) - \eta \Delta(w)$$

where $\Delta(w)$ is the error in estimating $\nabla f(w)$, we assume that $\|\Delta(w)\| \leq b$ for some $b > 0$. Next we show the expansiveness of $G_{f,\eta}$.

**Lemma 7.** *For any $\eta \leq 2/\beta$, we have*

$$\|G_{f,\eta}(w) - G_{f,\eta}(v)\| \leq \|w - v\| + 2\eta b$$

*Proof.*

$$
\begin{aligned}
&\|G_{f,\eta}(w) - G_{f,\eta}(v)\| \\
&= \|w - \eta \nabla f(w) - \eta \Delta(w) - v + \eta \nabla f(v) + \eta \Delta(v)\| \\
&\leq \|w - \eta \nabla f(w) - v + \eta \nabla f(v)\| + \eta(\|\Delta(w)\| + \|\Delta(v)\|) \\
&\leq \|w - v\| + \eta(\|\Delta(w)\| + \|\Delta(v)\|) \\
&\leq \|w - v\| + 2\eta b
\end{aligned}
$$

where the last equality holds because of the expansiveness of the conventional GD. $\square$

Now we can present the stability result of $\mathcal{A}_{\text{meta-NSGD}}$.

**Lemma 8.** *(Stability of $\mathcal{A}_{\text{meta-NSGD}}$) In $\mathcal{A}_{\text{meta-NSGD}}$, suppose $\eta \leq 2/\beta$, where $\beta$ is the smoothness parameter of $L_Z(h)$. Then $\mathcal{A}_{\text{meta-NSGD}}$ is $\alpha$-uniformly stable with $\alpha = L^2 \frac{T\eta}{K} + L\eta T b$.*

*Proof.* Consider $T$ steps of $\mathcal{A}_{\text{meta-NSGD}}$. Let $G_1, G_2 \ldots G_T$ be the noise vector, $\Delta_t = \frac{1}{m} \sum_{Z \in B_t} (\nabla L_Z(h_t) - \hat{\nabla} L_Z(h_t))$ for $t \in [T]$ and we have $\|\Delta_t\| \leq b$. We further denote the $\mathcal{I}_1, \ldots \mathcal{I}_T$ be the index sets of the mini-batch selected in $T$ iterations. Consider any pair of datasets $S$ and $S'$ that differ on $k_{\text{th}}$ element. Let $h_0, h_1, \ldots h_T$ and $h_0, h_1', \ldots h_T'$ denote the trajectories of $\mathcal{A}_{\text{meta-NSGD}}$ corresponding to $S$ and $S'$ respectively. Let $\xi_t = h_t - h_t'$.

The proof mainly follows from Lemma 3.4 in [19]. For any iteration $\tau$, we fix and the randomness of $G_\tau$ and $\mathcal{I}_\tau$. Denote $r$ be the number of occurrences of index $k$ in $\mathcal{I}_\tau$. By the expansiveness property of approximate gradient in Lemma 7, we have

$$\|\xi_{\tau+1}\| \leq \|\xi_\tau\| + 2L\eta \frac{r}{m} + 2\eta b$$

Then we can release the randomness of $G_\tau$ and $\mathcal{I}_\tau$, and note that $r$ is a Binomial random variable with mean $m/K$, then

$$\mathbf{E}\left[\|\xi_{\tau+1}\|\right] \leq \mathbf{E}\left[\|\xi_\tau\|\right] + \frac{2L\eta}{K} + 2\eta b$$

Since $\xi_0 = 0$, then $\mathbf{E}\left[\|\xi_\tau\|\right] \leq \frac{2L\eta\tau}{K} + 2\eta\tau b$

Then since $L_Z(h)$ is $L$-Lipshtiz, thus for any $Z$, we have

$$\mathbf{E}[L_Z(\bar{h}_T) - L_Z(\bar{h}_T')] \leq L\frac{1}{T}\sum_{t=1}^{T} \mathbf{E}[\|\xi_t\|]$$

$$\leq L\frac{1}{T}\sum_{t=1}^{T} \frac{2L\eta t}{K} + 2\eta t b$$

$$\leq L^2 \frac{\eta(T+1)}{K} + L\eta(T+1)b$$

which completes the proof.

$\square$

**Algorithm 3:** Approximate SGD

---

**input** : Initial iterate $w_0$, number of iterations $T$, parameter space $\mathcal{W}$

1 **for** $t = 1, \ldots, T$ **do**

2      choose $v_t$ such that $\mathbf{E}[v_t|w^{(t)}] = \nabla f(w^{(t)})$

3      update $w^{(t+1)} = \text{Proj}_{\mathcal{W}}\left(w^{(t)} - \eta(v_t + G_t + \Delta_t)\right)$ where $G_t \sim \mathcal{N}(0, \sigma^2 \mathbb{I}_d)$

4 **end**

5 where the $G_t$ is added Gaussian noise to preserve privacy and $\Delta_t$ is the error in estimating $v_t$.

**output** : $\bar{w} = \frac{1}{T} \sum_{t=1}^{T} w^{(t)}$

---

Next we show the convergence rate of SGD with approximate gradients. Consider an SGD algorithm on a differentiable convex function $f$,

**Lemma 9.** *Let $M, L, b > 0$. Let $f$ be a convex function, $\mathcal{W} = \mathcal{B}(0, M)$ and $w^* \in \arg\min_{w \in \mathcal{W}} f(w)$. Assume that for all $t$, $\|v_t\| \leq L$ and $\|\Delta_t\| \leq b$ with probability 1. Then*

$$\mathbf{E}[f(\bar{w})] - f(w^*) \leq \frac{M^2}{2\eta T} + \frac{\eta(L+b)^2}{2} + \eta\sigma^2 d + 2bM,$$

*where the expectation is taken over the randomness in $v_t, G_t$ and $\Delta_t$.*

*Proof.* Denote $\hat{v}_t = v_t + G_t + \Delta_t$, then by Lemma 14.1 in [29], we have

$$\underset{\hat{v}_{1:T}}{\mathbf{E}} \left[ \frac{1}{T} \sum_{t=1}^{T} \langle w^{(t)} - w^*, \hat{v}_t \rangle \right] \leq \frac{\|w^*\|^2}{2T\eta} + \frac{\eta}{2T} \sum_{t=1}^{T} \mathbf{E}\left[\|\hat{v}_t\|^2\right]$$

$$\leq \frac{M^2}{2\eta T} + \frac{\eta(L+b)^2}{2} + \eta\sigma^2 d$$

For any $t > 1$, we fix the randomness of $\hat{v}_{1:t-1}$. Since $w^{(t)}$ only depends on $\hat{v}_{1:t-1}$, then $w^{(t)}$ is fixed.

Therefore,

$$\underset{\hat{v}_t}{\mathbf{E}} \left[ \langle w^{(t)} - w^*, \hat{v}_t \rangle \right] = \langle w^{(t)} - w^*, \nabla f(w^{(t)}) + \mathbf{E}[\Delta_t] \rangle$$

$$= \langle w^{(t)} - w^*, \nabla f(w^{(t)}) \rangle + \langle w^{(t)} - w^*, \mathbf{E}[\Delta_t] \rangle$$

$$\geq \langle w^{(t)} - w^*, \nabla f(w^{(t)}) \rangle - 2bM$$

$$\geq f(w^t) - f(w^*) - 2bM$$

The second to the last step holds because $\|w^{(t)} - w^*\| \leq 2M$ and $\|\Delta_t\| \leq b$. The last step is due to the convexity of $f$.

By releasing the randomness of $\hat{v}_{1:t-1}$, summing over $t$ and dividing $T$, we have

$$\underset{\hat{v}_{1:T}}{\mathbf{E}} \left[ \frac{1}{T} \sum_{t=1}^{T} (f(w^{(t)}) - f(w^*)) \right] \leq \underset{\hat{v}_{1:T}}{\mathbf{E}} \left[ \frac{1}{T} \sum_{t=1}^{T} \langle w^{(t)} - w^*, \hat{v}_t \rangle \right] + 2bM$$

$$\leq \frac{M^2}{2\eta T} + \frac{\eta(L+b)^2}{2} + \eta\sigma^2 d + 2bM$$

Combining with the fact that $\mathbf{E}_{\hat{v}_{1:T}}[f(\bar{w}) - f(w^*)] \leq \mathbf{E}_{\hat{v}_{1:T}} \left[ \frac{1}{T} \sum_{t=1}^{T} (f(w^{(t)}) - f(w^*)) \right]$, we complete the proof. $\square$

## A.2 Proof of Theorem 4

*Proof.* Denote $\hat{\mathcal{E}}_n(h, \bar{Z})$ be the empirical counterpart of $\hat{\mathcal{E}}_n(h; \rho)$, which is written as

$$\hat{\mathcal{E}}_n(h, \bar{Z}) = \frac{1}{K} \sum_{Z \in \bar{Z}} L_Z(h)$$

Then we have

$$\mathbf{E}[\hat{\mathcal{E}}_n(\bar{h}_T;\rho)] - \min_{h\in\mathcal{W}}\hat{\mathcal{E}}_n(h;\rho) \le \mathbf{E}[\hat{\mathcal{E}}_n(\bar{h}_T,\bar{Z})] - \min_{h\in\mathcal{W}}\hat{\mathcal{E}}_n(h;\rho) + 4L^2\frac{\eta T}{K} + 2L\eta T b \tag{10}$$

$$\le E\left[\hat{\mathcal{E}}_n(\bar{h}_T,\bar{Z}) - \min_{h\in\mathcal{W}}\hat{\mathcal{E}}_n(h,\bar{Z})\right] + 4L^2\frac{\eta T}{K} + 2L\eta T b \tag{11}$$

$$\le \frac{M^2}{2\eta T} + \frac{\eta(2L+b)^2}{2} + \eta\sigma^2 d + 2bM + 4L^2\frac{\eta T}{K} + 2L\eta T b \tag{12}$$

where (10) is based on the stability result in Lemma 8, (11) follows from the fact that $\mathbf{E}\left[\min_{h\in\mathcal{W}}\hat{\mathcal{E}}_n(h,\bar{Z})\right] \le \min_{h\in\mathcal{W}}\mathbf{E}[\hat{\mathcal{E}}_n(h,\bar{Z})] = \min_{h\in\mathcal{W}}\hat{\mathcal{E}}_n(h;\rho)$. Finally, (12) is due to Lemma 9 on the convergence result of SGD with approximate gradient.

By plugging into the value of $T$, $\eta$ and $b$, we obtain the stated bound. $\qquad\square$

### A.3 Proof of Theorem 5

*Proof.* The excess risk defined in equation (2) can be decomposed into three parts:

$$\mathbf{E}[\mathcal{E}_n(\bar{h}_T;\rho)] - \mathcal{E}_\rho = A + B + C$$

where

$$
\begin{aligned}
A &= \mathbf{E}\left[\mathcal{E}_n(\bar{h}_T;\rho) - \hat{\mathcal{E}}_n(\bar{h}_T;\rho)\right] \\
B &= \mathbf{E}\left[\hat{\mathcal{E}}_n(\bar{h}_T;\rho)\right] - \hat{\mathcal{E}}_n(\bar{w};\rho) \\
C &= \hat{\mathcal{E}}_n(\bar{w};\rho) - \mathcal{E}_\rho
\end{aligned}
$$

To bound $A$, we fix the randomness of the task specific distribution $\mu$ and training output $h$, then

$$
\begin{aligned}
&\mathbf{E}_{Z\sim\mu^n}R_\mu(w_h(Z)) - \mathbf{E}_{Z\sim\mu^n}L_Z(h) \\
&= \mathbf{E}_{Z\sim\mu^n}R_\mu(w_h(Z)) - \mathbf{E}_{Z\sim\mu^n}\left[R_Z(w_h) + \frac{\lambda}{2}\|w_h - h\|^2\right] \\
&\le \mathbf{E}_{Z\sim\mu^n}\left(R_\mu(w_h(Z)) - R_Z(w_h(Z))\right) \\
&\le \frac{2L^2}{\lambda n}
\end{aligned}
$$

The first inequality holds since the norm is always non-negative, and the second inequality holds because $w_h(Z)$ is a minimizer of a regularized ERM thus $\frac{2L^2}{\lambda n}$-replace one stable. Then we release the randomness of $\mu$ and $h$ and obtain

$$A \le \frac{2L^2}{\lambda n}$$

To bound B, since $\bar{w}\in\mathcal{W}$, by Theorem 4, we have

$$
\begin{aligned}
B &= \mathbf{E}\left[\hat{\mathcal{E}}_n(\bar{h}_T)\right] - \hat{\mathcal{E}}_n(\bar{w}) \\
&\le \mathbf{E}\left[\hat{\mathcal{E}}_n(\bar{h}_T)\right] - \min_{h\in\mathcal{W}}\hat{\mathcal{E}}_n(h) \\
&= O\left(ML\cdot\left(\frac{1}{\sqrt{K}} + \frac{\sqrt{d\log(1/\delta)}}{\epsilon K}\right)\right)
\end{aligned}
$$

Finally to bound $C$, given the task $\mu$, we let $w_\mu = \text{Proj}_{W_\mu}(\bar{w})$, then we have

$$
\begin{aligned}
&\hat{\mathcal{E}}_n(\bar{w}) - \mathcal{E}_\rho \\
&= \mathbf{E}_{\mu\sim\rho}\mathbf{E}_{Z\sim\mu^n}\left[L_Z(\bar{w}) - \mathbf{E}_{\mu\sim\rho}\mathbf{E}_{Z\sim\mu^n}R_\mu(w_\mu)\right] \\
&= \mathbf{E}_{\mu\sim\rho}\mathbf{E}_{Z\sim\mu^n}\left[L_Z(\bar{w}) - R_\mu(w_\mu) - \frac{\lambda}{2}\|w_\mu - \bar{w}\|^2 + \frac{\lambda}{2}\|w_\mu - \bar{w}\|^2\right] \\
&= \mathbf{E}_{\mu\sim\rho}\mathbf{E}_{Z\sim\mu^n}\left[L_Z(\bar{w}) - R_Z(w_\mu) - \frac{\lambda}{2}\|w_\mu - \bar{w}\|^2 + \frac{\lambda}{2}\|w_\mu - \bar{w}\|^2\right] \qquad (13) \\
&\leq \frac{\lambda}{2}\mathbf{E}_{\mu\sim\rho}\|w_\mu - \bar{w}\|^2 \qquad (14) \\
&= \frac{\lambda}{2}V^2
\end{aligned}
$$

where equation (13) is due to the independence of $w_\mu$ and $Z$ conditioned on $\mu$, therefore $\mathbf{E}_{Z\sim\mu^n}R_\mu(w_\mu) = \mathbf{E}_{Z\sim\mu^n}R_Z(w_\mu)$. Equation (14) holds because $L_Z(\bar{w})$ minimizes $R_Z(w) + \frac{\lambda}{2}\|w - \bar{w}\|^2$.

By combining everything together, we have

$$
\mathbf{E}\left[\mathcal{E}_n(w_{\bar{h}_T})\right] - \mathcal{E}_\rho \leq \frac{2L^2}{\lambda n} + O\left(ML \cdot \left(\frac{1}{\sqrt{K}} + \frac{\sqrt{d\log(1/\delta)}}{\epsilon K}\right)\right) + \frac{\lambda}{2}V^2
$$

Let $\lambda = \frac{2L}{V}\sqrt{\frac{1}{n}}$, the overall bound becomes

$$
\frac{2LV}{\sqrt{n}} + O\left(ML \cdot \left(\frac{1}{\sqrt{K}} + \frac{\sqrt{d\log(1/\delta)}}{\epsilon K}\right)\right)
$$

$\square$

## B   Communication-efficient Meta-learning with DP-FTMRL

In this section, we present the DP-FTMRL algorithm and the guarantee it provides.

---

**Algorithm 4:** $\mathcal{A}_{\text{DP-FTMRL}}$: Differentially Private Follow-the-Meta-Regularized-Leader

**input** : Collection of datasets $\bar{Z} = [Z_1, \ldots Z_K]$ arriving in a stream, noise variance $\sigma^2$, regularization parameter $\lambda_{\text{meta}}$, approximation error $b$

1  $h_1 \leftarrow \arg\min_{h\in\mathcal{W}} \lambda_{\text{meta}}\|h\|^2$. **output** : $h_1$
2  Server do: $\mathcal{T} \leftarrow \text{InitializeTree}(n, \sigma^2, 2L)$
3  **for** $t = 1, \ldots, K-1$ **do**
4  $\quad$ Server sends $h_t$ to task $t$.
5  $\quad$ **for** *task t* **do**
6  $\quad\quad$ Solve $R_{Z_t,\lambda}(w; h_t)$ and give an estimate $\hat{w}_{h_t}(Z_t)$ such that $\|\hat{w}_{h_t}(Z_t) - w_{h_t}(Z_t)\| \leq \frac{b}{\lambda}$
7  $\quad\quad$ Send $\hat{\nabla}L_{Z_t}(h_t) = -\lambda(\hat{w}_{h_t}(Z_t) - h_t)$ back to server
8  $\quad$ **end**
9  $\quad$ Server adds $\hat{\nabla}L_{Z_t}(h_t)$ to $\mathcal{T}$: $\mathcal{T} \leftarrow \text{AddToTree}(\mathcal{T}, t, \hat{\nabla}L_{Z_t}(h_t))$
10 $\quad$ Server aggregates the gradients: $s_t \leftarrow \text{GetSum}(\mathcal{T}, t)$
11 $\quad$ Server computes $h_{t+1} = \arg\min_{h\in\mathcal{W}}\langle s_t, h\rangle + \frac{\lambda_m}{2}\|h\|^2$ **output** : $h_{t+1}$
12 **end**
$\quad$ **output** : $\bar{h}_K = \frac{1}{K}\sum_{t=1}^K h_t$

---

where the functions InitializeTree, AddToTree and GetSum are associated with the tree aggregation protocol. Roughly speaking, InitializeTree initializes the tree structure $\mathcal{T}$ and AddtoTree adds the gradient $\hat{L}_{Z_t}(h_t)$ in to $\mathcal{T}$. GetSum returns the sum of $\{\nabla\hat{L}_{Z_i}(h_i)\}_{i=1}^t$ privately, and the returned value can be expressed as $\sum_{i=1}^t \nabla\hat{L}_{Z_i}(h_i) + b_t$ where $g_t$ is a zero mean Gaussian random variable. More details of these functions can be found in Appendix B of [23].

**Theorem 10.** *(privacy guarantee) Let $\epsilon \leq 2\ln(1/\delta)$. Under Assumption 1, let $[h_1, \ldots h_K]$ and $\bar{h}_K$ be the outputs of Algorithm 4. By setting $\sigma = \frac{2\lceil\log(n+1)\rceil\ln(1/\delta)}{\epsilon}$ and $b = \min\left\{L, \frac{L\sigma\sqrt{d\lceil\log(n+1)\rceil\ln(n)}}{K}\right\}$, Algorithm 4 guarantees $(\epsilon, \delta)$ task-level differential privacy.*

Note that since $b \leq L$, then we have $\left\|\hat{\nabla}L_{Z_t}(h_t)\right\| \leq \|\nabla L_{Z_t}(h_t)\| + b \leq 2L$. Thus, the proof of the privacy guarantee follows similar lines to that of [23, Theorem 4.1].

We measure the performance of $\mathcal{A}_{\text{DP-FTMRL}}$ by regrets against any $w^* \in \mathcal{W}$ defined as

$$R_{\bar{Z}}(\mathcal{A}_{\text{DP-FTMRL}}; w^*) = \frac{1}{K}\sum_{t=1}^{K} L_{Z_t}(h_t) - \frac{1}{K}\sum_{t=1}^{K} L_{Z_t}(w^*)$$

We rely on the following lemma to provide a regret guarantee of DP-FTMRL, which is stated as follows

**Lemma 11.** *([30, Lemma 7]) Let $\phi_1 : \mathcal{C} \to \mathbb{R}$ be a convex function s.t. $\theta_1 \in \arg\min_{\theta \in \mathcal{C}} \phi_1(\theta)$ exists. Let $\Psi(\theta)$ be a convex function s.t. $\phi_2(\theta) = \phi_1(\theta) + \Psi(\theta)$ is 1-strongly convex w.r.t. $\|\cdot\|$-norm. Let $\theta_2 \in \arg\min_{\theta \in \mathcal{C}} \phi_2(\theta)$. Then for any $b$ in the subgradient of $\Psi$ at $\theta_1$, the following is true: $\|\theta_1 - \theta_2\|_* \leq \|b\|_*$. Here $\|\cdot\|_*$ is the dual norm of $\|\cdot\|$.*

**Theorem 12.** *(Regret guarantee) Recall the settings in Theorem 10. Let $\{h_1, \ldots h_K\}$ be the outputs of $\mathcal{A}_{\text{DP-FTMRL}}$. Then, for any $w^* \in \mathcal{W}$, w.p. at least $1 - \beta$ over the algorithm's randomness, we have*

$$R(\mathcal{A}_{\text{DP-FTMRL}}; w^*) = O\left(LM \cdot \left(\frac{1}{\sqrt{K}} + \sqrt{\frac{d^{1/2}\ln^2(1/\delta)\ln(1/\beta)}{\epsilon n}}\right)\right)$$

The proof follows similar lines to the proof of [23, Theorem 5.1] with a slight modification, where we take into account the effect of the gradient approximation error. The proof is provided here for completeness.

*Proof.* By the end of iteration $t$, we have

$$h_{t+1} = \arg\min_{h \in \mathcal{W}}\langle s_t, h\rangle + \frac{\lambda_m}{2}\|h\|^2 + \langle b_t, h\rangle$$

$$= \arg\min_{h \in \mathcal{W}}\sum_{i=1}^{t}\langle\hat{\nabla}L_{Z_i}(h_i), h\rangle + \frac{\lambda_m}{2}\|h\|^2 + \langle b_t, h\rangle$$

$$= \arg\min_{h \in \mathcal{W}}\sum_{i=1}^{t}\langle\nabla L_{Z_i}(h_i) + \Delta_i, h\rangle + \frac{\lambda_m}{2}\|h\|^2 + \langle b_t, h\rangle$$

$$= \arg\min_{h \in \mathcal{W}}\sum_{i=1}^{t}\langle\nabla L_{Z_i}(h_i), h\rangle + \frac{\lambda_m}{2}\|h\|^2 + \langle b_t + \sum_{i=1}^{t}\Delta_i, h\rangle$$

where $\Delta_i$ is the approximation error for $\hat{\nabla}L_{Z_i}(h_i)$. Denote $e_t = b_t + \sum_{i=1}^{t}\Delta_i$, since we have $\|\Delta_i\| \leq b \leq \frac{L\sigma\sqrt{d\lceil\log(n+1)\rceil\ln(n)}}{K}$ and $\|b_t\| \leq L\sigma\sqrt{d\lceil\log(n+1)\rceil\ln(n/\beta)}$ with probability $1 - \beta$. Therefore, with probability over $1 - \beta$, the norm of $n_t$ is bounded by

$$\|e_t\| \leq \sum_{i=1}^{t}\|\Delta_i\| + \|b_t\| = O\left(L\sigma\sqrt{d\lceil\log(n+1)\rceil\ln(n/\beta)}\right)$$

Therefore, by denoting the non-private objective as $\tilde{h}_t = \arg\min_{h \in \mathcal{W}}\sum_{i=1}^{t}\langle\nabla L_{Z_i}(h_i), h\rangle + \frac{\lambda_m}{2}\|h\|^2$. By Lemma 11, we can bound $\left\|\tilde{h}_{t+1} - h_{t+1}\right\|$ by

$$\left\|\tilde{h}_{t+1} - h_{t+1}\right\| \leq \frac{\|e_t\|}{\lambda}$$

Finally, we can bound the regret as

$$R_{\bar{Z}}(\mathcal{A}_{\text{DP-FTMRL}}; w^*) = \frac{1}{K}\sum_{t=1}^{K} L_{Z_t}(h_t) - \frac{1}{K}\sum_{t=1}^{K} L_{Z_t}(w^*)$$

$$\leq \frac{1}{K}\sum_{t=1}^{K}\langle \nabla L_{Z_t}(h_t), h_t - h^*\rangle$$

$$= \frac{1}{K}\sum_{t=1}^{K}\langle \nabla L_{Z_t}(h_t), h_t - \tilde{h}_t + \tilde{h}_t - h^*\rangle$$

$$= \frac{1}{K}\sum_{t=1}^{K}\langle \nabla L_{Z_t}(h_t), \tilde{h}_t - h^*\rangle + \frac{1}{K}\sum_{t=1}^{K}\langle \nabla L_{Z_t}(h_t), h_t - \tilde{h}_t\rangle \quad (15)$$

The first term of (15) can be bounded by $\left(\frac{L^2}{\lambda} + \frac{\lambda}{2n}\left(\|h^*\|^2 - \|h_1\|^2\right)\right)$ [[31], Theorem 5.2]. The second term can be bounded by the concentration bound of $\|e_t\|$, such that

$$\frac{1}{K}\sum_{t=1}^{K}\langle \nabla L_{Z_t}(h_t), h_t - \tilde{h}_t\rangle \leq \frac{1}{K}\sum_{t=1}^{K} L\left\|\tilde{h}_{t+1} - h_{t+1}\right\| \leq O\left(\frac{L\sigma\sqrt{d\lceil\log(n+1)\rceil\ln(n/\beta)}}{\lambda}\right)$$

By setting $\lambda$ optimally and plugging into the value of $\sigma$, we complete the proof. $\qquad\square$

The utility guarantee of $\mathcal{A}_{\text{DP-FTMRL}}$ is therefore derived via online-to-batch conversion [32].

**Theorem 13.** *(Utility guarantee) Recall the settings in Theorem 10. With probability $1 - \beta$ over the algorithm's randomness, we have*

$$\mathbf{E}[\hat{\mathcal{E}}(\bar{h}_K; \rho)] - \min_{h\in\mathcal{W}}\hat{\mathcal{E}}(h; \rho) \leq O\left(LM\cdot\left(\frac{1}{\sqrt{K}} + \sqrt{\frac{d^{1/2}\ln^2(1/\delta)\ln(1/\beta)}{\epsilon n}}\right)\right)$$

Finally we can obtain the transfer risk in a similar way as in Section 4, which is stated as follows:

**Theorem 14.** *(Transfer risk) Recall the settings in Theorem 10. We have*

$$\mathbf{E}[\mathcal{E}_n(\bar{h}_K; \rho)] - \mathcal{E}_\rho \leq \frac{2L^2}{\lambda n} + \frac{\lambda}{2}V^2 + O\left(LM\cdot\left(\frac{1}{\sqrt{K}} + \sqrt{\frac{d^{1/2}\ln^2(1/\delta)\ln(1/\beta)}{\epsilon n}}\right)\right)$$

*By letting $\lambda = \frac{2L}{V}\sqrt{\frac{1}{n}}$, we have*

$$\mathbf{E}[\mathcal{E}_n(\bar{h}_K; \rho)] - \mathcal{E}_\rho \leq \frac{2LV}{\sqrt{n}} + O\left(LM\cdot\left(\frac{1}{\sqrt{K}} + \sqrt{\frac{d^{1/2}\ln^2(1/\delta)\ln(1/\beta)}{\epsilon n}}\right)\right)$$

*where $\bar{h}_K$ is the output of $\mathcal{A}_{DP\text{-}FTMRL}$ and the expectation is taken w.r.t. $\bar{Z}$ and the inner randomness of the algorithm.*

## C  Additional Experiments

We evaluate our single- and multi-cluster meta learning algorithms on the Omniglot [28] benchmark dataset for few-shot classification tasks. Omniglot contains 1623 characters (i.e., 1623 classes) and each character contains 20 ($28 \times 28$) black and white images. This benchmark dataset is split into training and testing data as follows: the training dataset contains all data samples from 1461 characters while the testing dataset contains the data points of the remaining 162 characters. The training dataset is used in the meta training to obtain the meta models, and the testing dataset is used to evaluate the accuracy of the base learner (for a given task) with respect to the learned meta models.

In our experiments, we set the total number of tasks (users) $K = 50000$. For any given task, the goal is to perform an $M$-shot $N$-way classification. In an $M$-shot $N$-way classification, we sample $N$

classes ($N$ characters) uniformly from the testing classes (i.e., out of 162 characters) and subsequently sample $M + 1$ data points from each class. Among these $M + 1$ data points, $M$ of them are used for training and the remaining one is for testing. We use the same network structure as in [1, 2, 4] with 4 convolutional modules. The code is partly based on the public repo in [33].

We focus on the 5-shot 5-way classification setting. We use SGD as the base learner for the single cluster algorithm and Adam as the base learner for the multi-cluster algorithm. We follow similar hyperparameter settings for the step size and the number of iterations for the base learner as in [2]. In the meta training phase of our algorithms, we sample $K = 50000$ few-shot classifications tasks as our training task datasets. We set the number of iterations for the meta training as $T = 20000$ and the training shots (the number of training data points per class) as 10. We set the regularization parameter $\lambda = 0.1$, the clip norm $\hat{L} = 0.5$, the batch size $= 25$, the privacy parameter $\epsilon \in \{3, 10, \infty\}$, where $\epsilon = \infty$ refers to the non-private setting, and the number of clusters $q \in \{1, 2\}$. That is, we report the accuracy resulting from different settings of the privacy parameter (corresponding to different values of the noise variance) in both the single-cluster ($q = 1$) and the 2-cluster ($q = 2$) cases. We report results with $95\%$ confidence interval.

Table 1: Accuracy results on Omniglot few-shot classification task, $q$ is the number of meta models and $\epsilon$ is the privacy parameters

| $(q, \epsilon)$ | 5-shot 5-way |
| --- | --- |
| $(1, 3)$ | $75 \pm 1.7\%$ |
| $(1, 10)$ | $85.4 \pm 1.4\%$ |
| $(1, \infty)$ | $94.2 \pm 0.9\%$ |
| $(2, 3)$ | $78.5 \pm 2.1\%$ |
| $(2, 10)$ | $79.3 \pm 1.8\%$ |
| $(2, \infty)$ | $85.4 \pm 2\%$ |

Compared to the non-private baseline ($\epsilon = \infty$), our single-cluster private algorithm has about $8.8\%$ accuracy drop for $\epsilon = 10$. This accuracy loss increases to $19.2\%$ with higher privacy ($\epsilon = 3$). Similar accuracy gaps of $6.1\%$ for $\epsilon = 10$ and $6.9\%$ for $\epsilon = 3$ are also reported in the multi-cluster algorithm. We note that the multi-cluster algorithm does not lead to any noticeable improvement in accuracy compared to the single-cluster algorithm. One explanation for that is that the underlying task distribution of the Omniglot benchmark seems to be well represented by a single meta model, and hence, using the clustering-based algorithm does not necessarily yield an improved performance (let alone the fact that the algorithm will also have a slower convergence to a good solution). This is indeed what we observe in our experiments for the 2-cluster setting. In particular, in these experiments, we observed that almost all the users eventually fall into one cluster as the algorithm converges. For example, in the setting where $q = 2$ and $\epsilon = 3$, our results showed that $97.7\%$ of the sampled users (tasks) eventually chose the same meta model to update.

In [8], the authors report accuracy around $65\%$ for the 5-shot 5-way Omniglot few-shot classification with $\epsilon = 9.5$ and $10^6$ sampled tasks. Our algorithm attains a noticeable improvement in accuracy ($85.4\%$) with comparable privacy guarantee ($\epsilon = 10$) and only $5 \times 10^4$ sampled tasks. Even with stronger privacy guarantee ($\epsilon = 3$), our algorithms still achieve better performance ($75\%$ for single-cluster training and $78.5\%$ for 2-cluster training). Moreover, we provide task-level differential privacy guarantee while [8] provides record-level differential privacy.