# OpenReview forum: "Task-level Differentially Private Meta Learning"
_NeurIPS.cc/2022/Conference — NeurIPS 2022 Accept_

### Official Review · Reviewer_GS79 · 2022-07-08

**Rating:** 4
**Confidence:** 4
**Soundness:** 2 fair
**Presentation:** 2 fair
**Contribution:** 2 fair

**Summary:**

In this work, the authors propose meta learning algorithms with task-level differential privacy to protect the entire dataset for each task. The authors also conduct experiments to empirically evaluate the effectiveness of our proposed algorithms.

**Questions:**

[1] Can the proposed algorithm be applied to other losses, e.g., the general convex algorithms?

[2] The influence of the task difficulty and the meta models’ weights on the task similarity is not studied. Theoretical and empirical analysis should be given.

[3] It is not clear about the derivation of the set of optimal parameters for the tasks in Eqn. (8). The definition of optimality is undefined in the paper. In addition, the difference comparison between $V_{c}$ and $V$ is not given.

[4] It would be better if the authors conduct experiments on the large-scale real-world datasets to evaluate the performance of the proposed methods.

[5] More discussions on the personalized privacy-preserving requirements of different tasks should be given.

**Limitations:**

Yes

**Strengths And Weaknesses:**

Pros

[+] The authors propose a task-level differentially private meta learning algorithm based on the meta-regularization approach.

[+] The authors propose a private clustering based meta learning algorithm to enable private meta learning of multiple meta models.



Cons

[-] The contribution of this paper seems limited. For example, in this paper, the authors directly apply existing differential privacy (DP) optimization techniques like Noisy SGD (e.g., [1]) on the newly defined "data point" and loss function achieves task-level differential privacy naturally. In addition, the proposed meta-clustering algorithm is directly derived from the HYP CLUSTER algorithm in [2].

[-] The authors make strong assumptions, e.g., the assumption of the strongly convexity of the base learner $R_{Z,\lambda}$.

[-] The experiment evaluation is limited. Firstly, there are some existing DP meta learning algorithms and task-level DP algorithms, and the authors fail to compare the proposed algorithms with those baselines. In addition, the influence of different task distributions is unexplored in the experiments. Lastly, the size of the adopted dataset is small.

[1] Bassily, Raef, Vitaly Feldman, Kunal Talwar, and Abhradeep Guha Thakurta. "Private stochastic convex optimization with optimal rates." Advances in neural information processing systems 32 (2019).

[2] Mansour, Yishay, Mehryar Mohri, Jae Ro, and Ananda Theertha Suresh. "Three approaches for personalization with applications to federated learning." arXiv preprint arXiv:2002.10619 (2020).

---

> ### Author Response · Authors · 2022-08-02
> **Response to Reviewer GS79**
>
> Thanks for the valuable feedback. Please, see our response to your comments and questions below.
>
> -  Regarding the comment about the limited contribution of our work, we respectfully disagree with the reviewer. First, our proposed method is not a straightforward application of DP optimization techniques to existing meta-learning results. There are crucial differences between our approach and the existing methods based on meta-regularization. Please, refer to the second point of our response to the first reviewer (Reviewer zJxt) for a detailed explanation. Second, although our meta-clustering algorithm has a similar outline to the HYP-CLUSTER algorithm, we find this similarity to be natural as both algorithms are inspired by standard clustering techniques. Moreover, HYP-CLUSTER has a different objective and operates under a different problem setting from ours. For example, in HYP-CLUSTER, users from the same cluster share the same model while in our algorithm users from the same cluster can still learn their own model using their base learner.
>
> -  We disagree with the claim that our work entails making strong assumptions. The assumptions we make in this work are fairly standard and, in fact, weaker than those invoked in many prior works on meta-learning. Please, refer to point 3 of our response to the first reviewer (Reviewer zJxt) for more details. Also, please note that the input loss in our work is *only* assumed to be convex and Lipschitz. We do not assume strong convexity in this work. The quantity $R_{Z,\lambda}$, mentioned in your review, is strongly convex due to regularization ($R_{Z,\lambda}$ is a regularized objective constructed inside the algorithm).
>
>
> -  Regarding the comment on the experimental evaluation, we did present additional experiments of the Omniglot few-shot classification benchmark in the appendix where we show improved performance over existing DP meta learning algorithms even though we provide stronger (task-level) DP guarantees.
>
> We now answer the questions from the reviewer:
>
> 1. Our results apply to general convex (and Lipschitz) losses. As noted above, we do not assume strong convexity.
> 2. We measure the task difficulty by the $L_2$ norm of model parameters and assume it is always bounded by some quantity $M$ (See Assumption 1, line 142). There is no direct relationship between $M$ and the task similarity $V$. Intuitively, one can view the task similarity as the “variance” of the task model parameters. It is possible that the parameter for each single task has a large norm while the parameters from all tasks are concentrated and close to each other, thus having a small variance.
> 3. The definition of the set of optimal parameters $W_\mu$ is given in line 130. $W_\mu$ is the set of parameters that minimizes the population risk for task $\mu$. As for the difference between $V_c$ and $V$, both can be seen as a measure of the variation of the model parameters. $V$ is the variation among all tasks, while $V_c$ offers more flexibility by allowing partitioning the tasks into $q$ groups first and computing the average of the variation for each group. It is easy to show that $V_c$ is no larger than $V$. When the meta distribution contains several well-separated concentrations,  $V_c$ can be much smaller than $V$.
> 4. We present additional experiments on the Omniglot few-shot classification benchmark in the appendix as noted earlier in this response.
> 5. Studying the setting where different tasks have different privacy requirements is an interesting research direction, but this is orthogonal to the problem we address here. In this work, we focus on a more fundamental goal, namely, developing task-level DP meta learning algorithms with strong theoretical guarantees on their utility.

---

### Official Review · Reviewer_awko · 2022-07-10

**Rating:** 6
**Confidence:** 3
**Soundness:** 3 good
**Presentation:** 3 good
**Contribution:** 3 good

**Summary:**

Meta learning has been applied successfully in the Machine learning area recently to generalize new task with new small datasets efficiently including meta-initialization and meta-regularization methods. Meanwhile, meta learning may suffer from leakage of sensitive information and privacy part of meta learning has not been well studied. Although a work with meta-initialization method designs a record-level differential privacy approach, it is still too weak to be against aggregate statistics and has potential leakage as well as unrealistic (strict) assumption. To fill the gaps, this work proposes a meta-regularization-based task-level DP algorithm and guarantee transfer risk as well as accuracy with less strict assumption.

**Questions:**

1. Since the manuscript mentioned meta learning in the distributed setting, did you consider secure multiparty computation as another feasible approach to tackle privacy issue during meta learning?

**Limitations:**

No potential negative societal impact found.

**Strengths And Weaknesses:**

Strength:
1. Assumption of this work is less strict as existing works (only assume convex and Lipschitz loss).
2. Existing DP optimization (Noisy SGD) can be applied to this algorithm as almost “free lunch”.
3. Design a novel task-level DP clustering-based meta learning algorithm for multiple meta models as well.

Weakness:
1. Like existing work based on meta-initialization only, this work is based on meta-regularization only. Since this work has mentioned those two methods could be connected by framework of online convex optimization, it would be nice to consider meta-initialization as well under the mentioned framework.

---

> ### Author Response · Authors · 2022-08-02
> **Response to Reviewer awko**
>
> Thank you for the thoughtful review of our work.
>
> -  Regarding the comment about the meta-initialization approach, we would like to clarify that although there is a connection between meta-initialization and meta-regularization (namely, the base learner objective in meta-initialization can be viewed as a first-order approximation of the objective in meta-regularization), such a connection is not strong enough to provide a recipe for devising variants of our algorithms based on the meta-initialization approach. Devising private algorithms based on meta-initialization seems to be quite challenging since this approach generally requires solving a non-convex optimization problem even when the loss function is convex. Existing works get around this issue by making more assumptions about the loss function (e.g., quadratic growth property or strong convexity), which we do not make in this work.
>
> -  As for the question concerning secure multiparty computation, we believe this is an interesting research direction. In this work, we focus on providing statistical data privacy guarantees based on differential privacy, which hides the contribution of each user in the aggregate computations performed by the distributed algorithm. Secure MPC can offer an extra level of protection by hiding the intermediate model iterates. However, combining secure MPC with DP will likely involve much more communication and computational cost. Addressing these issues is an interesting direction for future work.

---

### Official Review · Reviewer_zJxt · 2022-07-11

**Rating:** 5
**Confidence:** 3
**Soundness:** 3 good
**Presentation:** 3 good
**Contribution:** 2 fair

**Summary:**

This paper proposes a task-level differentially private meta-learning algorithm based on the meta-regularization method. The authors provide the privacy and utility guarantees based on convex and Lipschitz loss assumptions. They also present a clustering-based method to deal with the case where the tasks are sampled from multi-modal distributions. They evaluate the performance of the algorithms in the case of linear regression tasks with mean square loss.

**Questions:**

none

**Limitations:**

Yes

**Strengths And Weaknesses:**

1.	This paper studies the problem of meta-learning with task-level differential privacy. However, the motivation of this paper is not clear. The authors claim that the record-level is not sufficient in many scenarios (line 9, 53) and they present an “email spam filter” example (line 53-56) to support this point. However, they do not clearly explain what kind of “sensitive information” (line 56) can be protected by task-level DP instead of record-level DP, which makes the example unconvincing and the paper not well-motivated.

2.	The proposed approach is conceptually simple. They apply noisy SGD [14] to meta-level updates of the meta-regularization method [3,4] to get their task-level private meta-learning algorithm. The analysis of privacy and utility guarantees also follows [14], which is quite standard.

3.	The assumptions in Assumption 1 are too strong for meta-learning scenarios. Meta-learning methods are typically applied in reinforcement learning and deep learning scenarios [1-6], which generally do not satisfy Assumption 1. As the privacy guarantee analysis of the proposed algorithms relies on assumption 1, the proposed algorithms are not practical in meta-learning scenarios.

4.	The experiments demonstrate the effectiveness of the proposed methods. However, the experiments are also very toyish (only in the case of linear regression task with MSE loss on toy dataset), which reduces the practical values. This could be also because of the overly strong assumptions (as mentioned in the third review).

---

> ### Author Response · Authors · 2022-08-02
> **Response to Reviewer zJxt**
>
> We would like to thank the reviewer for the valuable feedback, please see the corresponding response below.
>
> 1. Task-level differential privacy is a strictly stronger notion than record-level privacy. With task-level DP, the privacy protection is guaranteed for the entire dataset of each task rather than for a single record as in record-level DP. In our email spam filter example, the frequency of commonly used words like company name is considered as aggregate information thus not protected by record-level DP. Another example is a user’s language preference, which is typically a common information shared across all email records in a user’s dataset, and hence, will not be protected by record-level DP either. Since such common/aggregate information can still be exploited to reveal a user's identity, we believe that developing task-level DP meta learning algorithms with strong theoretical guarantees is well-motivated.
> 2. Our proposed approach is not a straightforward application of DP optimization methods to existing works in the meta-learning literature such as [3, 4]. There are some crucial differences between our approach and the existing methods in the non-private meta-regularization literature. For example, in [3], the authors only consider linear prediction models. Meanwhile, the statistical analysis in [4] assumes that the optimal biased vector is already learned and does not formally study the dependence of the excess transfer risk on the number of tasks and the task similarity. We address these issues by formulating a meta objective suitable for private learning and provide formal bounds on the excess transfer risk when the loss is convex and Lipschitz.
> 3. Assuming convex and Lipschitz loss is fairly standard in works that offer formal utility guarantees on the transfer risk. In fact, our assumption is weaker than many existing meta-learning works. For example, references [7, 8] in our paper assume the loss to satisfy the quadratic growth property while [Finn et al. 2019] requires strong convexity for their analysis. We are not aware of any prior work that provides formal transfer risk guarantees with more relaxed assumptions than ours. However, we do believe it is interesting to extend our results to non-convex regimes that cover broader meta-learning scenarios.
> 4. We present additional experiments on the Omniglot few-shot classification benchmark in the appendix. There, we show noticeable improvements over prior DP meta learning methods even though we provide stronger privacy guarantees.
>
> [Finn et al. 2019] Chelsea Finn, Aravind Rajeswaran, Sham Kakade, and Sergey Levine. "Online meta-learning." In International Conference on Machine Learning, pp. 1920-1930. PMLR, 2019.

---

### Meta-Review · Area_Chair_tt8K · 2022-08-23

**Recommendation:** Accept
**Confidence:** Less certain

**Metareview:**

The paper resulted in lukewarm support for the paper, although they were positive. There were some concerns regarding the practicality of the algorithms mentioned in the paper. Also, I took a quick look at the paper. The setup seems similar to model personalization (https://papers.nips.cc/paper/2018/hash/aa97d584861474f4097cf13ccb5325da-Abstract.html), where folks have analyzed non-convex models with DP. The paper misses citing this line of work. I will request the authors to include the discussions from the rebuttal phase, and also add a discussion to compare with the work along the lines of DP personalization (mentioned above).

**Award:**

No

---

### Decision · Program_Chairs · 2022-09-14

Accept